# Attenuating Effects of Pyrogallol-Phloroglucinol-6,6-Bieckol on Vascular Smooth Muscle Cell Phenotype Changes to Osteoblastic Cells and Vascular Calcification Induced by High Fat Diet

**DOI:** 10.3390/nu12092777

**Published:** 2020-09-11

**Authors:** Myeongjoo Son, Seyeon Oh, Ji Tae Jang, Chul-Hyun Park, Kuk Hui Son, Kyunghee Byun

**Affiliations:** 1Department of Anatomy & Cell Biology, Gachon University College of Medicine, Incheon 21936, Korea; mjson@gachon.ac.kr; 2Functional Cellular Networks Laboratory, College of Medicine, Department of Medicine, Graduate School and Lee Gil Ya Cancer and Diabetes Institute, Gachon University, Incheon 21999, Korea; seyeon8965@gmail.com; 3Aqua Green Technology Co., Ltd., Smart Bldg., Jeju Science Park, Cheomdan-ro, Jeju 63309, Korea; whiteyasi@gmail.com; 4Department of Thoracic and Cardiovascular Surgery, Gachon University Gil Medical Center, Gachon University, Incheon 21565, Korea; cdgpch@gilhospital.com

**Keywords:** receptor of AGEs, Toll-like receptor 4, *Ecklonia cava* extract, pyrogallol-phloroglucinol-6,6-bieckol, vascular calcification

## Abstract

Advanced glycation end products/receptor for AGEs (AGEs/RAGEs) or Toll like receptor 4 (TLR4) induce vascular smooth muscle cell (VSMC) phenotype changes in osteoblast-like cells and vascular calcification. We analyzed the effect of *Ecklonia cava* extract (ECE) or pyrogallol-phloroglucinol-6,6-bieckol (PPB) on VSMC phenotype changes and vascular calcification prompted by a high-fat diet (HFD). HFD unregulated RAGE, TLR4, transforming growth factor beta (TGFβ), bone morphogenetic protein 2 (BMP2), protein kinase C (PKC), and nuclear factor kappa-light-chain-enhancer of activated B cells (NF-κB) signals in the aorta of mice. ECE and PPB restored the increase of those signal pathways. AGE- or palmitate-treated VSMC indicated similar changes with the animal. HFD increased osteoblast-like VSMC, which was evaluated by measuring core-binding factor alpha-1 (CBFα-1) and osteocalcin expression and alkaline phosphatase (ALP) activity in the aorta. ECE and PPB reduced vascular calcification, which was analyzed by the calcium deposition ratio, and Alizarin red S stain was increased by HFD. PPB and ECE reduced systolic, diastolic, and mean blood pressure, which increased by HFD. PPB and ECE reduced the phenotype changes of VSMC to osteoblast-like cells and vascular calcification and therefore lowered the blood pressure.

## 1. Introduction

Vascular calcification, which is defined as calcified deposits in media or intima, leads to medial calcification of the vessel or atherosclerotic intima calcification [1]. Vascular calcification is increased by age and aggravated in chronic diseases including diabetes and chronic kidney disease, atherosclerosis, dyslipidemia, hypertension, and bone-mineral disorders [2,3,4,5]. It is also a risk factor of the morbidity and mortality prompted by cardiovascular disease [6]. Vascular calcification is a process that is similar with the bone formation process [1] and is initiated by phenotype change of vascular smooth muscle cells (VSMCs) to osteoblast-like cells, followed by deposition of hydroxyapatite [7]. Physiologically normal VSMCs maintain blood flow by regulating the contraction and relaxation of vessels [8]. Nonetheless, VSMCs from atherosclerotic regions lost their physiological characteristics, which regulate contraction or relaxation [8]. The vessel that has osteoblast-like cells demonstrated an increased vascular stiffness, which leads to reducing blood flow [8]. The osteoblast-like VSMCs secreted bone-related protein biomarkers involving osteogenesis such as osteoblast differentiation and maturation [7,8,9]. Bone morphogenetic protein-2 (BMP2) is increased during vascular calcification and leads to the activation of core-binding factor alpha-1 (CBFα-1), a principal transcriptional regulator of the maturation of osteoblasts in the bone [10,11,12]. CBFα-1 also increased the expression of osteoblast proteins in VSMCs and led to phenotype changes to osteoblast-like cells [13]. Alkaline phosphatase (ALP) is known to be an early marker of osteoblast activity, while osteopontin and osteocalcin are elevated in the late calcification process [14,15]. Those proteins have an important role in the formation and deposition of hydroxyapatite [15]. Those bioactive molecules are elevated during vascular calcification. It is well known that the upregulation of advanced glycation end products (AGEs) and their receptors (RAGEs) involved in VSMC phenotype changes to osteoblast-like cells. By AGE binding to RAGE, inflammation-related signaling cascades such as nuclear factor (NF)-κB, extracellular signal-regulated kinase (ERK) 1/2, mitogen-activated protein kinases (p38 MAPK), c-Jun N-terminal kinases (JNK) [16,17], and transforming growth factor β (TGF β) [18] are activated. Those signal pathways lead to the upregulation of CBFα-1 and VSMC phenotype changes to osteoblast-like cells [19]. TLR4/NF-κB is a classic signaling pathway that involves inflammation [20] and is also related to the calcification of VSMCs [21]. It is known that obesity or a high-fat diet (HFD) increases the expression of RAGE or RAGE ligand. Obese subjects had a higher RAGE expression in adipocytes than lean subjects [22]. The accumulation of RAGE ligands is also elevated in adipose tissues of HFD-induced obesity animal models [23,24,25]. TLR4 and NF-κB expressions were also elevated in adipose tissues of HFD-induced obesity models [26]. Ecklonia cava (E. cava) is a species of brown alga that demonstrated anti-inflammatory, anti-oxidative, anti-diabetic, and anti-obesity effects [27,28,29]. Our earlier study demonstrated that E. cava extract (ECE) had an attenuation effect on increased expressions of RAGE, TLR4, and NF-κB in adipose tissue by HFD [26]. Pyrogallol-phloroglucinol-6,6-bieckol (PPB), a compound of E. cava, was reported to show a decreased vascular inflammation by decreasing monocyte-associated endothelial cell death and reducing monocyte involved in VSMC proliferation or migration [30]. Nonetheless, no study has assessed the effect of ECE or PPB on VSMC phenotype change to osteoblast-like cell or vascular calcification. Therefore, we assessed the effect of ECE and PPB on vascular calcification, which is induced by HFD through decreasing AGE/RAGE pathway and TLR4/NF-κB pathway in VSMC.

## 2. Materials and Methods

Detailed experimental methods are available in the Appendix A.

### 2.1. HFD-Fed Mice Model

C57BL/6N mice (male, eight weeks of age) were bought from Orient bio (Seongnam, Korea) and kept at a constant temperature of approximately 23 °C, relative humidity of 50%, and a light/dark cycle of 12 h/12 h. The mice were divided into six groups (five mice/group):

(1st group) Normal fat diet (NFD)-fed mice group: mice received NFD for four weeks and then 0.9% normal saline orally administered with NFD for the last four weeks.

(2nd group) HFD-fed mice group: mice received 45% HFD (Research Diet, New Brunswick, NJ, USA) for four weeks and then 0.9% normal saline orally administered with HFD for the last four weeks.

(3–5th groups) HFD-fed mice orally administered with ECE group: mice received 45% HFD for four weeks and then ECE orally administered with HFD for the last four weeks. The ECE groups were divided according to ECE dose concentration (3rd group, 50 mg/kg/day; 4th group, 100 mg/kg/day; or 5th group, 150 mg/kg/day).

(6th group) HFD-fed mice orally administered with PPB group; mice received 45% HFD for four weeks and then PPB orally administered with HFD for the last four weeks (2.5 mg/kg/day).

At the end of the eight week study period, all mice were sacrificed in accordance with the ethical principles of the Institutional Animal Care and Use Committee of Gachon University (approval number: LCDI-2019-0097).

### 2.2. Preparation of ECE and Isolation of PPB

ECE was acquired from Aqua Green Technology Co., Ltd. (Jeju, Korea). For extraction, E. cava were thoroughly washed and air-dried at room temperature for two days, the leaves were ground, and 50% alcohol was added, followed by incubation at 85 °C for 12 h. The ECE were filtered and then, concentrated, sterilized by heating to high temperature (over 85 °C) for 40 to 60 min, and then spray-dried. The PPB was then isolated following a prior study [30]. Briefly, centrifugal partition chromatography was conducted using a two-phase solvent system composed of water, ethyl acetate, methyl alcohol, and n-hexane mixture (ratio 7:7:3:2). The organic stationary phase was filled in the chromatography column, followed by pumping of the mobile phase into the column in a descending manner at the same flow rate utilized for separation (2 mL/min). We finally validated that the purity of the PPB of around 91.24% was utilized in the study [30].

### 2.3. Vascular Smooth Muscle Cell In Vitro Modeling

#### 2.3.1. Cultivation

Mouse vascular aortic smooth muscle cells (MOVAS) were acquired from American Type Culture Collection and grown with Dulbecco’s modified Eagle’s medium (Welgene, Gyeongsan, Korea), which is supplemented with 10% fetal bovine serum and 0.2 mg/mL G-418 anti-biotics (Gibco, Grand Island, NY, USA).

#### 2.3.2. Preparation of Palmitic Acid Conjugation to Bovine Serum Albumin

To allow palmitic acid conjugation to bovine serum albumin (palmitate (PA), Sigma–Aldrich, St. Louis, MO, USA), 2.267 g of fatty acid-free bovine serum albumin (BSA; Sigma–Aldrich, St. Louis, MO, USA) was thawed in 100 mL of pre-warmed 150 mM sodium chloride. The mixed solution was then stirred until it is completely dissolved in a water bath at 37 °C. While the BSA was being stirred, 30.6 mg of sodium palmitate (Sigma–Aldrich, St. Louis, MO, USA) was thawed in 50 mL of 150 mM sodium chloride in a water bath at 70 °C. The palmitate solution was then split into 10 mL portions, 5 mL of which was transferred to the BSA solution, and then the solution was stirred at 37 °C for 1 h, adjusted to a final volume to 100 mL with 150 mM sodium chloride, and then adjusted pH to 7.4 with 1M sodium hydroxide.

#### 2.3.3. In Vitro Modeling

To create the palmitate-treated group, 200 μM palmitate with or without 5, 25, or 50 μg/mL ECE or 1.8 μg/mL PPB was treated to MOVAS for 24 h, and then the cell was washed with phosphate-buffered saline (PBS) thrice. For another way to create the AGE-treated group, 800 ng/mL AGE with or without 5, 25, or 50 μg/mL ECE or 1.8 μg/mL PPB was treated to MOVAS for 24 h, and then the cell was washed with PBS a third time. After all procedures were finished, each experiment begun and all experiments were repeated at least three times.

### 2.4. Immunohistochemistry (3,3-Diaminobenzidine; DAB)

Blocks of paraffin-embedded aorta tissue were sectioned to 10 µm thickness, placed on a coating slide, and dried at 37 °C for 24 h. The slides were then deparaffinized with xylene and incubated with 0.3% hydrogen peroxide (Sigma–Aldrich, St. Louis, MO, USA) for 30 min. Afterward, slides were rinsed twice with PBS and incubated in normal animal serum to reduce non-specific antibody-antigen binding and then incubated with anti-RAGE (Santa Cruz Biotechnology, Dallas, TX, USA; dilution rate 1:200), anti-NF-kB (Cell Signaling, Danvers, MA, USA; dilution rate 1:250), or anti-TLR4 antibody (Novus Biologicals, Centennial, CO, USA; dilution rate 1:200) at 4 °C for two days, followed by three additional rinses with PBS. Slides were then treated with biotinylated secondary antibodies from the ABC kit (Vector Laboratories, San Francisco, CA, USA; dilution rate 1:100), which was incubated for an hour with the blocking solution and rinsed thrice with PBS. Slides were left to react with 3,3′-diaminobenzidine substrates for 15 min, and they were mounted with a cover slip and DPX mounting solution (Sigma–Aldrich, St. Louis, MO, USA). Images were seen with the use of a light microscope (Olympus, Tokyo, Japan), and the quantification of the intensity of the brown color (arrow) was made with the use of the Image J software 1.53 version (NIH, Bethesda, MD, USA).

### 2.5. RNA Extraction and Quantitative Real-Time Polymerase Chain Reaction (qrt-PCR)

To extract RNA from cell and aorta tissues, RNAiso Plus (TAKARA Bio, Kyoto, Japan) was used according to the instruction manual. Resuspend the pellets with 1 mL of RNAiso Plus mixed in 0.1 mL of chloroform (Amresco, Cleveland, OH, USA), and then centrifuged at 12,000× *g* for 15 min at 4 °C. The clear part (supernatant) was mixed with 0.25 mL absolute isopropanol, and isolated RNA pellets were washed with 70% alcohol and centrifuged at 7500× *g* for 5 min at 4 °C. Dried pellets were dissolved in 5 to 30 μL of diethyl pyrocarbonate (DEPC)-treated water and extract RNA was quantified using a Nanodrop 2000 (Thermo Fisher Scientific, Waltham, MA, USA). Appropriate primers listed in Appendix A, the forward primers, reverse primers, distilled water, template complementary DNA (cDNA), and SYBR green (TAKARA Bio, Kyoto, Japan) mixed and placed in a 384-well plate. The mixed samples were validated using a PCR machine (Bio-Rad, Berkeley, CA, USA).

### 2.6. Enzyme-Linked Immunosorbent Assay (ELISA)

To confirm AGE level in serum, an aliquot of the withdrawn blood (0.5 to 0.7 mL) was added in serum separator tubes (Becton Dickinson, Franklin Lakes, NJ, USA) and then, centrifuged at 2000× *g* for 20 min. After completion, the separated serum specimens were moved into a new tube and stored in a freezer at −80 °C. The AGE antibody (1 µg/mL) was coated in a 96-well plate with 100 mM bicarbonate/carbonate buffer (pH 9.6) at 4 °C overnight. Following the removal of the antibody, 5% skim milk (Sigma–Aldrich) was incubated at 4 °C overnight. After washing with PBS, the serum specimens were added in the plate, followed by incubation for 2 h at 4 °C overnight. The plate was washed with PBS, and then the peroxidase-conjugated secondary antibody incubated for 2 h at room temperature (Vector Laboratories, San Francisco, CA, USA). To develop the level of AGE, 3,3′,5,5′-tetramethylbenzidine solution (Sigma–Aldrich, St. Louis, MO, USA) was added for development, followed by incubation for 8 min and mixed with an equal volume of 2N sulfuric acid, and the optical density was read at 450 nm.

### 2.7. Histological Hematoxylin and Eosin (H & E) Staining

Blocks of paraffin-embedded aorta tissue were sectioned to a thickness of 10 µm, placed on a coating slide, and dried at 37 °C for 24 h. Slides were deparaffinized with xylene and alcohol and incubated in Mayer’s hematoxylin (DAKO, Carpinteria, CA, USA) for 1 min, eosin (Sigma–Aldrich, St. Louis, MO, USA) for 20 s, followed by three rinses with distilled water. Finally, slides were mounted with the DPX solution (Sigma–Aldrich, St. Louis, MO, USA), followed by detection with a light microscope (Olympus, Tokyo, Japan). The intima and media thicknesses of mice aorta were measured with the use of the image J software.

### 2.8. Blood Pressure Measurement

Blood pressures was measured with the use of a noninvasive tail-cuff CODA system (Kent Scientific Corp., Torrington, CT, USA). The mice sublimation was performed for 7 min for three days, 10 min for two days, and 13 min for two days per group, and the systolic, diastolic, and mean artery blood pressures were measured in all animals prior to sacrifice.

### 2.9. Statistical Analysis

The Kruskal–Wallis and Mann–Whitney U post-hoc tests determined the significance of differences among the NFD (PBS), HFD (AGE or PA), HFD/ECE50 (AGE/ECE5 or PA/ECE5), HFD/ECE100 (AGE/ECE25 or PA/ECE25), HFD/ECE150 (AGE/ECE50 or PA/ECE50), and HFD/PPB (AGE/PPB or PA/PPB) groups. Results are presented as mean ± standard deviation, and *p*-values of <0.05 were considered statistically significant. Means denoted by a different letter indicate significant differences between groups. The analysis was conducted with the use of the SPSS version 22 (IBM Corporation, New York, NY, USA). The asterisk (*) indicates difference between some groups vs. NFD (or PBS-treated cell) group and the sharp (#) indicates difference between some groups vs. HFD (or AGE or PA-treated cell) group.

## 3. Results

### 3.1. ECE and PPB Reduced the Expression of RAGE and TLR4 Increased by HFD in the Aorta

The C57BL/6N mice were fed with NFD or HFD for eight weeks and some parameters were validated (Appendix A). The serum level of AGE for HFD-fed mice was higher than in NFD-fed mice, and it was reduced by either PPB or ECE. The reducing effect was most prominent on 150 mg/kg of ECE and PPB administration (Figure 1A). The expression of RAGE in the aorta of HFD-fed mice was increased, and it was reduced by either ECE or PPB administration (Figure 1B). There was no significant difference in the decreasing effect among 50, 100, and 150 mg/kg of ECE and PPB administration. The expression of TLR4 in the aorta of HFD-fed mice was elevated, and it was reduced by PPB or ECE administration (Figure 1C). The reducing effect on TLR4 expression among 100 and 150 mg/kg of ECE and PPB was not significantly different.

### 3.2. ECE and PPB Attenuated the Expression of RAGE and TLR4 by Either AGE or Palmitate

The expression of RAGE increased by treating either AGE or palmitate to MOVAS, and it was reduced by PPB or ECE (Figure 2A,B). The reducing effect was most prominent in the 25 and 50 μg/mL of ECE- or PPB-treated MOVAS. The expression of TLR4 was increased by treating either AGE or palmitate to MOVAS, and it was reduced by PPB or ECE (Figure 2C,D). The decreasing effect was most prominent in the 25 and 50 μg/mL of ECE- or PPB-treated MOVAS.

### 3.3. ECE and PPB Reduced the Expression of PKC, NOX2, TGFβ, BMP2, and NF-κB by HFD or by Either AGE or Palmitate

The expressions of protein kinase C (PKC), TGFβ, BMP2, and NF-κB in the MOVAS were elevated by either AGE or palmitate. Those expressions were reduced by treating either ECE or PPB. The reducing effect in the 25 and 50 μg/mL of ECE or PPB were not significantly different (Figure 3 and Appendix A).

The increased expressions of PKC, TGFβ, BMP2, and NF-κB were found in the aorta by HFD, and those were reduced by either PPB or ECE administration. The reducing effect on PKC and TGFβ were most prominent in the PPB-administered mice aorta. The reducing effect on BMP2 was not significantly different between 50 mg/kg of ECE or PPB administration (Figure 3 and Appendix A). The reducing effect on NF-κB was not significantly different among 50, 100, and 150 mg/kg of ECE and PPB administration (Figure 3 and Appendix A). The expression of nicotinamide adenine dinucleotide phosphate (NADPH) oxidase 2 (NOX2) was increased by HFD and it was decreased by either of ECE or PPB administration (Appendix A). The decreasing effect was most prominent in 150 mg/kg of ECE and PPB administration.

### 3.4. ECE and PPB Reduced VSMC Phenotype Changes to Osteoblast-Like Cell and Vascular Calcification

The expression of CBFα1 was elevated by treating either AGE or palmitate, and it was reduced by the addition of PPB or ECE (Figure 4A,B). The reducing effect was not significantly different in both 25 and 50 μg/mL of ECE and PPB. The expression of CBFα1 was elevated by HFD, and it was reduced by either PPB or ECE. The reducing effect was most prominent in PPB administration (Figure 4C).

The expression of osteocalcin was elevated by either AGE or palmitate, and this was reduced by ECE or PPB. The reducing effect was not significantly different in both 25 and 50 μg/mL of ECE and PPB. The expression of osteocalcin was elevated by HFD in the aorta, and it was reduced by ECE or PPB. The reducing effect was not significantly different in both 100 and 150 mg/kg of ECE and PPB administration (Figure 4D–F). ALP activity was elevated in the aorta of HFD mice than NFD-fed mice. ALP activity was reduced by ECE or PPB. The reducing effect was not significantly different in both 100 and 150 mg/kg of ECE and PPB administration (Figure 4G).

The calcium deposition ratio in the aorta was elevated by HFD, and it was restored by ECE or PPB administration. The reducing effect was not significantly different in both 100 and 150 mg/kg of ECE and PPB administration (Figure 4H). The calcium deposition ratio in the AGE- or palmitate-treated MOVAS was elevated, and those were reduced by the addition of ECE or PPB. The reducing effect was not significantly different in both 25 and 50 μg/mL of ECE and PPB (Appendix A). The positively stained cells by Alizarin red S in the aorta was increased by HFD, and it was reduced by ECE or PPB (Figure 4I).

### 3.5. ECE and PPB Reduced the Blood Pressure and Intima-Media Ratio, Which Is Elevated by HFD

The systolic, diastolic, and mean blood pressures were elevated by HFD, and those were reduced by treating ECE or PPB. The decreasing effect was not significantly different among 50, 100, and 150 mg/kg of ECE and PPB administration (Figure 5A–C). The intima-media ratio was reduced by HFD, and it was elevated by treating ECE or PPB administration. The increasing effect was not significantly different among 50, 100, and 150 mg/kg of ECE and PPB administration (Figure 5D–E). 

## 4. Discussion

It is well known that the AGE/RAGE pathway is involved in vascular calcification by the enhancement of VSMC phenotype changes to osteoblast-like cells. AGE-treated VSMCs exhibited an increased expression of CBFα-1 mRNA, ALP activity, and osteocalcin secretion [31]. AGEs are synthesized by a reaction between a lysine or a hydroxylysine of a protein and sugar [32]. Therefore, diabetes-induced hyperglycemia is a well-known condition that increases the formation of AGEs [33]. HFD also elevated the AGE deposition. Apart from that, HFD elevated the expressions of RAGE and TLR4 in adipose tissue [26]. In our study, HFD increased the level of serum AGE and expressions of RAGE and TLR4 in mice aorta. It seems that HFD could increase the expressions of RAGE and TLR4 in not only the adipose tissue but also the aorta. Both ECE and PPB reduced the serum level of AGE and expressions of RAGE and TLR4 in the aorta, which was elevated by HFD. We created an in vitro model of HFD by treating palmitate to mouse aortic VSMCs. By treating palmitate, the RAGE and TLR4 expressions were elevated in VSMC, and those were reduced by the addition of either ECE or PPB. AGE-treated VSMC also lead to elevated expressions of RAGE and TLR4. Those expressions were reduced by either ECE or PPB. Although the knock-down experiments of TLR4 and RAGE were not conducted, we showed that ECE and PPB can inhibit vascular calcification through TLR4 and RAGE through this study.

For the downstream signals of RAGE, PKC/MAPK/ERK1/2 activates CBFα-1 [19]. PKC activates Reactive oxygen species (ROS) generating NOX2 [34], and the NOX2 might be involve in vascular calcification [35,36]. Upregulated NOX2 induced increased expression of CBFα-1 and led to calcification of vascular interstitial cells [37].

AGE also activated TGFβ and BMP2, which in turn activated Smad signals and led to an increased expression of CBFα-1 [19]. NF-kB was activated by both RAGE and TLR4 [19]. It is well known that RAGE/NF-κB pathway leads to the upregulation of CBFα-1 [38]. Earlier studies indicated that AGE treatment resulted in the upregulation of NF-κB in VSMC and led to vascular calcification [38,39]. In our study, NF-κB, PKC, TGFβ, and BMP2, which prompted the upregulation of CBFα-1 in MOVAS, were increased by either AGE or palmitate treatment. Those increases were reduced by the addition of ECE or PPB. In the aorta of HFD-fed mice, NF-κB, PKC, TGFβ, and BMP2 were higher than in the NFD-fed mice. Those increases were reduced by ECE or PPB in the animal (Figure 3 and Appendix A). It is well known that increased CBFα-1 in VSMCs leads to phenotype changes to osteoblast-like cells [13,40]. Our study demonstrated that the expression of CBFα-1 in VSMCs was elevated by either AGE or palmitate treatment, and it was reduced by treating ECE or PPB. In addition to that, the expression of CBFα-1 in the aorta was elevated by HFD, and it was reduced by ECE of PPB (Figure 4).

We calculated the calcium deposit ratio in the aorta of mice. The HFD-increased calcium deposition ratio in the aorta and treatment of PPB or ECE lead to a decrease of the calcium deposition ratio. Alizarin red S stain exhibited a similar trend with the calcium ratio. The number of cells that positively stained with Alizarin red S were elevated by HFD in the aorta of mice, and it was reduced by PPB or ECE. The calcium deposition ratio of MOVAS was elevated by treating either AGE or palmitate, and it was reduced by the addition of PPB or ECE in the MOVAS (Figure 4).

During the osteogenesis process, ALP is chiefly involved in cleaving pyrophosphate to phosphate and promoting hydroxyapatite deposition and leads to bone mineralization [41]. Osteocalcin had a role in the regulation of hydroxyapatite size and shape [42] and increased during the calcification process [14,15]. Many studies indicated that ALP and osteocalcin were elevated during VSMC phenotype changes to osteoblast-like cells [40,43]. In our study, HFD elevated the ALP activity and the expression of osteocalcin in the aorta of mice, and those were reduced by treating ECE or PPB (Figure 4). VSMC phenotype changes are associated with developing systolic hypertension, increasing intimal thickening, increasing arterial contraction, decreasing relaxation, and inducing arterial stiffness [44].

In our study, the systolic, diastolic, and mean blood pressures were elevated by HFD, and those are reduced by either PPB or ECE. The intima-media ratio was reduced in the aorta of HFD-fed mice, and it was elevated by either PPB or ECE (Figure 5A–C). It is proposed that the HFD induced the increased medial thickness of the aorta, and it was reduced by treating PPB or ECE. Our results indicated that HFD induced RAGE and TLR4 activation, which leads to the upregulation of TGFβ, BMP2, PKC, and NF-κB signals in the aorta of mice (Figure 3 and Appendix A). AGE or palmitate treatment in the MOVAS also elevated RAGE and TLR4 expressions and resulted in the upregulation of TGFβ, BMP2, PKC, and NF-κB signals. HFD increased the osteoblast-like VSMCs in the aorta of mice, which expressed CBFα-1 and osteocalcin and manifested an increased ALP activity in the aorta. Vascular calcification, which was evaluated using the calcium deposition ratio and Alizarin red S stain, was increased by HFD. ECE and PPB reduced osteoblast-like VSMCs and vascular calcification in the aorta. PPB and ECE reduced systolic, diastolic, and mean blood pressures, which increased due to HFD. PPB and ECE reduced the phenotype changes to osteoblast-like VSMCs and vascular calcification and therefore decreased the blood pressure (Figure 5F). The limitation of our study is that we used MOVAS instead of human origin VSMC, thus the effect of ECE or PPB on reducing osteoblast-like VSMCs and vascular calcification might be different between human and mice VSMC. In future studies, we should evaluate the of ECE or PPB effect on reducing osteoblast-like VSMCs with human VSMCs for human application. In addition, VSMC could be affected by different underlying comorbidities, it might be important which underlying mechanisms of VSMC calcification in primary isolated human VSMCs from donors of different age groups and with different underlying comorbidities [45].

## Figures and Tables

**Figure 1 nutrients-12-02777-f001:**
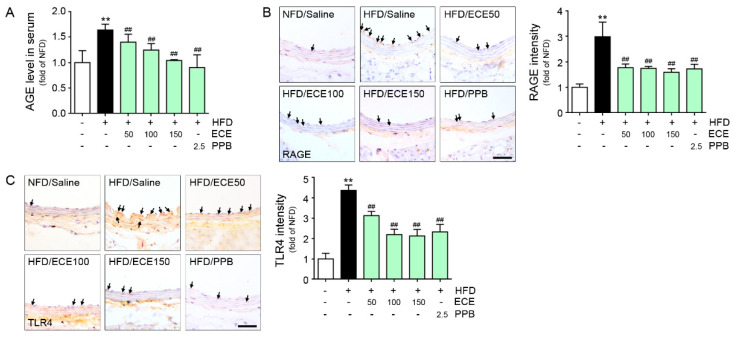
Comparative analysis of concentration-dependent ECE and PPB administration on the reduction of AGE and its receptor expression level in HFD-fed mice. (**A**) AGE levels were calculated in HFD-fed mice with or without ECE/PPB administration. (**B**) Light microscopic images showing RAGE expression (brown, arrow) in the aorta of HFD-fed mice. Quantitative graphs showing RAGE intensity from representative images. Scale bar = 50 μm. (**C**) Light microscopic images showing TLR4 expression (brown, arrow) in the aorta of HFD-fed mice. Quantitative graphs showing TLR4 intensity from representative images. Scale bar = 50 μm. **, *p* < 0.01 vs. NFD/Saline group; ##, *p* < 0.01 vs. HFD/Saline group. AGE, advanced glycation end products; ECE, *Ecklonia cava* extract; HFD, high-fat diet; NFD, normal fat diet; PPB, pyrogallol-phloroglucinol-6,6-bieckol; RAGE, receptor for advanced glycation end products; TLR4, toll-like receptor 4.

**Figure 2 nutrients-12-02777-f002:**
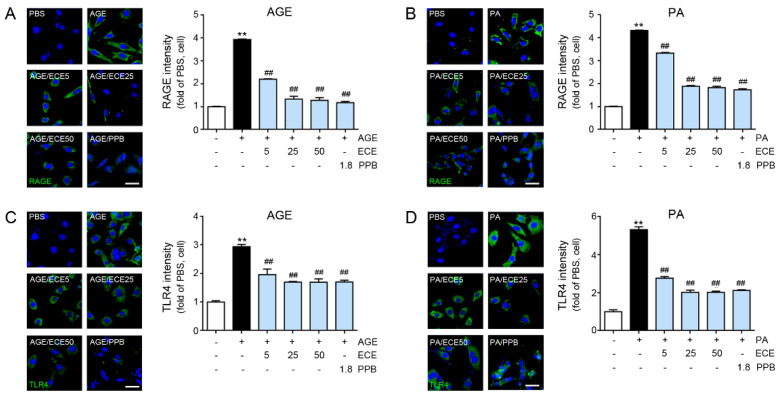
Comparative analysis of concentration-dependent ECE and PPB treatment on the reduction of RAGE and TLR4 receptor expression level in AGE or PA-treated VSMC. (**A**,**B**) Confocal fluorescence microscopic images demonstrating RAGE expression (green) and nuclei (DAPI, blue) in AGE or PA-treated VSMC. Quantitative graphs demonstrating RAGE intensity from representative images. Scale bar = 50 μm. (**C**,**D**) Confocal fluorescence microscopic images demonstrating TLR4 expression (green) and nuclei (DAPI, blue) in AGE or PA-treated VSMC. Quantitative graphs demonstrating TLR4 intensity from representative images. Scale bar = 50 μm; **, *p* < 0.01 vs. PBS group; ##, *p* < 0.01 vs. AGE or PA group. AGE, advanced glycation end products; DAPI, 4′,6-diamidino-2-phenylindole; ECE, *Ecklonia cava* extract; PA, palmitate; PBS, phosphate-buffered saline; PPB, pyrogallol-phloroglucinol-6,6-bieckol; RAGE, receptor for advanced glycation end products; TLR4, toll-like receptor 4; VSMC, vascular smooth muscle cell.

**Figure 3 nutrients-12-02777-f003:**
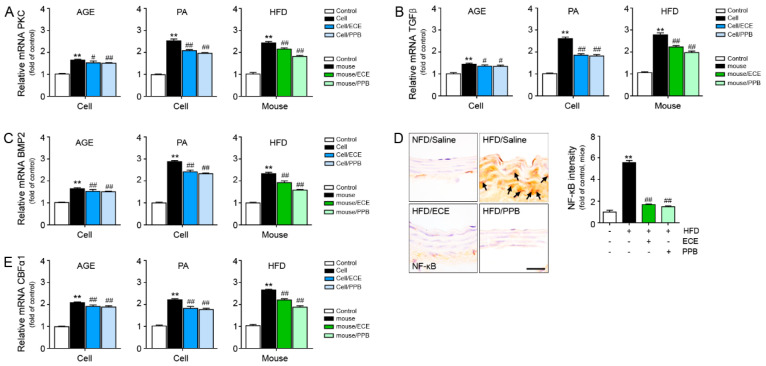
Comparative analysis of ECE and PPB treatment on the reduction of vascular smooth muscle cell phenotypic switching to osteoblast-like cell in AGE or palmitate-treated VSMC and HFD-fed mice. mRNA levels of the phenotypic switching to osteoblast-like cell-related molecules (**A**) PKC, (**B**) TGFβ, (**C**) BMP2, and (**E**) CBFα1 were determined using qRT-PCR. (**D**) Arrows indicate NF-κB protein expression of HFD-fed mice and quantified graph showing intensity. We incubated 25 μg/mL ECE or 1.8 μg/mL PPB with AGE or PA-treated VSMC (cell) and 100 mg/kg ECE or 2.5 mg/kg PPB administered with HFD-fed mice. scale bar = 25 μm. **, *p* < 0.01 vs. PBS or NFD/Saline group; #, *p* < 0.05, ##, *p* < 0.01 vs. AGE, PA or HFD/Saline group. AGE, advanced glycation end products; BMP2, Bone morphogenetic protein 2; CBFα1, core-binding factor alpha 1; ECE, *Ecklonia cava* extract; HFD, high-fat diet; NFD, normal fat diet; NF-κB, nuclear factor kappa-light-chain-enhancer of activated B cells; PA, palmitate; PBS, phosphate-buffered saline; PKC, protein kinase C; PPB, pyrogallol-phloroglucinol-6,6-bieckol; qRT-PCR, real-time quantitative reverse transcription polymerase chain reaction; RAGE, receptor for advanced glycation end products; TGFβ, transforming growth factor beta; VSMC, vascular smooth muscle cell.

**Figure 4 nutrients-12-02777-f004:**
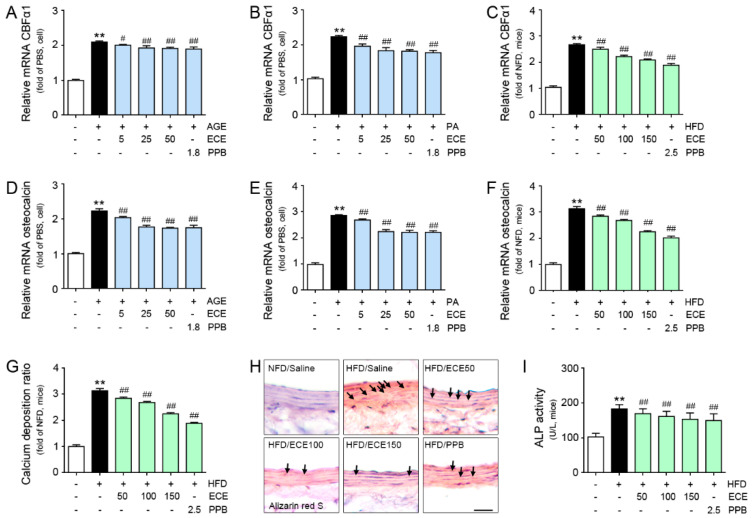
Comparative analysis of ECE and PPB treatment on the reduction of osteoblast-like cell marker expression in AGE or palmitate-treated VSMC and HFD-fed mice. (**A**–**C**) mRNA levels of the osteoblast-like markers CBFα1 in AGE- or PA-treated VSMC (cell) and HFD-fed mice were determined by qRT-PCR. (**D**–**F**) mRNA levels of the osteoblast-like markers osteocalcin in AGE- or PA-treated VSMC (cell) and HFD-fed mice were determined by qRT-PCR. (**G**) Calcium deposition was calculated using the calcium deposition assay and (**H**) Alizarin red S staining in HFD-fed mice. Positive signal (red dot, arrow) was marked with an arrow in images. Scale bar = 50 μm. (**I**) ALP activity was measured with the use of the aorta of HFD-fed mice. **, *p* < 0.01 vs. PBS or NFD/Saline group; #, *p* < 0.05, ##, *p* < 0.01 vs. AGE, PA or HFD/Saline group. AGE, advanced glycation end products; ALP, alkaline phosphatase; CBFα1, core-binding factor alpha 1; ECE, *Ecklonia cava* extract; HFD, high-fat diet; NFD, normal fat diet; PA, palmitate; PBS, phosphate-buffered saline; PPB, pyrogallol-phloroglucinol-6,6-bieckol; qRT-PCR, real-time quantitative reverse transcription polymerase chain reaction; VSMC, vascular smooth muscle cell.

**Figure 5 nutrients-12-02777-f005:**
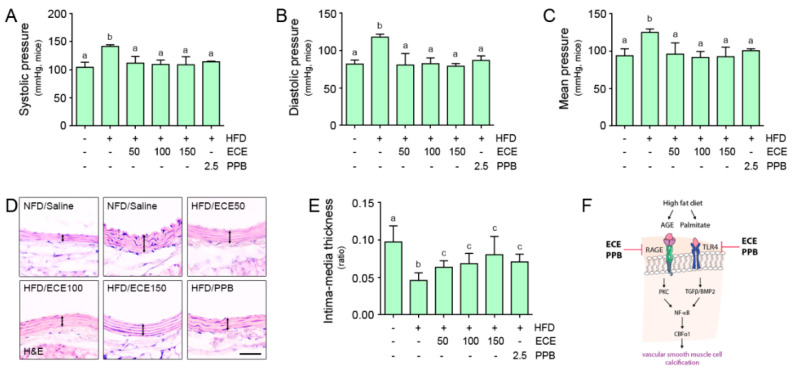
Comparative analysis of ECE and PPB administration on the regulation of blood pressure in HFD-fed mice. (**A**–**C**) Systolic, diastolic, and mean artery pressures were measured prior to sacrifice. (**D**) Light microscopic images showing H&E stained blood vessels and (**E**) Intima-media thickness acquired using representative H&E images. Arrows indicate media thickness. Scale bar = 50 μm. (**F**) Summary illustration image showing inhibitory effects of ECE and PPB on vascular smooth muscle cell calcification in HFD condition. Means denoted by a different letter indicate significant differences between groups. AGE, advanced glycation end products; BMP2, bone morphogenetic protein 2; CBFα1, core-binding factor alpha 1; ECE, *Ecklonia cava* extract; HFD, high-fat diet; H&E, hematoxylin and eosin stain; NFD, normal fat diet; PPB, pyrogallol-phloroglucinol-6,6-bieckol; NF-κB, nuclear factor kappa-light-chain-enhancer of activated B cells; RAGE, receptor for advanced glycation end products; TGFβ, transforming growth factor beta; TLR4, toll like receptor 4.

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
