# Peer review of "Attenuating Effects of Pyrogallol-Phloroglucinol-6,6-Bieckol on Vascular Smooth Muscle Cell Phenotype Changes to Osteoblastic Cells and Vascular Calcification Induced by High Fat Diet"

_nutrients, 2020, doi:10.3390/nu12092777_

Round 1

Reviewer 1 Report

Dear authors

This paper examines the effects of a novel compound ECE and its derivative PPB on VSMC phenotype.  It uses both in vivo and in vitro methods to suggest a consistent effect of these compounds, proposed to be via its effect on AGE and TLR4 mediated pathways.

Whilst the effects are modest, they appear consistent and a variety of techniques are used to demonstrate this.

Comments:

  • There is no figure 3
  • there is a no treatment control for these experiments, but there is no vehicle control.  Can the authors comment on the feasibility of including these?
  • Whilst the pathway for effect of ECE and PPB is proposed via RAGE and TLR4, there is no knockdown experiment to conclusively show this

Author Response

Response to Reviewer 1 Comments

This paper examines the effects of a novel compound ECE and its derivative PPB on VSMC phenotype.  It uses both in vivo and in vitro methods to suggest a consistent effect of these compounds, proposed to be via its effect on AGE and TLR4 mediated pathways.

Whilst the effects are modest, they appear consistent and a variety of techniques are used to demonstrate this.

Point 1: There is no figure 3

Response 1: We apologizes for this comment. We added the figure 3 in the manuscript.

Figure 3

Point 2: there is a no treatment control for these experiments, but there is no vehicle control. Can the authors comment on the feasibility of including these?

Response 2: We apologizes for any confusion you. The control group we used the vehicle control animals with oral administration of 0.9% normal saline in the normal fat diet (NFD)-fed mice. In addition, the saline for an accurate comparison to high fat diet (HFD)-fed mice were administered orally. To reduce the confusion of readers and reviewers, we have corrected this information in the material and methods section of manuscript.

Material and methods (previous version)

2.1. HFD-fed mice model

C57BL/6N mice (male, 8 weeks of age) were bought from Orient bio (Seongnam, Republic of Korea) and kept at a constant temperature of approximately 23°C, relative humidity of 50%, and a light/dark cycle of 12-hour/12-hour. The mice were divided into six groups (5 mice/group):

(1st group) Normal fat diet (NFD)-fed mice group: mice received NFD for 8 weeks.

(2nd group) HFD-fed mice group: mice received 45% HFD (research diet, USA) for 8 weeks.

(3-5th groups) HFD-fed mice orally administered with ECE group: mice received 45% HFD for 4 weeks and then ECE orally administered with HFD for the last 4 weeks. The ECE groups were divided according to ECE dose concentration (3rd group; 50 mg/kg/day, 4th group; 100 mg/kg/day or 5th group; 150 mg/kg/day).

(6th group) HFD-fed mice orally administered with PPB group; mice received 45% HFD for 4 weeks and then PPB orally administered with HFD for the last 4 weeks (2.5 mg/kg/day).

At the end of the 8-week study period, all mice were sacrificed in accordance with the ethical principles of the Institutional Animal Care and Use Committee of Gachon University (approval number: LCDI-2019-0097).

Material and methods (new version)

2.1. HFD-fed mice model

C57BL/6N mice (male, 8 weeks of age) were bought from Orient bio (Seongnam, Republic of Korea) and kept at a constant temperature of approximately 23°C, relative humidity of 50%, and a light/dark cycle of 12-hour/12-hour. The mice were divided into six groups (5 mice/group):

(1st group) Normal fat diet (NFD)-fed mice group: mice received NFD for 4 weeks and then 0.9% normal saline orally administered with NFD for the last 4 weeks.

(2nd group) HFD-fed mice group: mice received 45% HFD (research diet, USA) for 4 weeks and then 0.9% normal saline orally administered with HFD for the last 4 weeks.

(3-5th groups) HFD-fed mice orally administered with ECE group: mice received 45% HFD for 4 weeks and then ECE orally administered with HFD for the last 4 weeks. The ECE groups were divided according to ECE dose concentration (3rd group; 50 mg/kg/day, 4th group; 100 mg/kg/day or 5th group; 150 mg/kg/day).

(6th group) HFD-fed mice orally administered with PPB group; mice received 45% HFD for 4 weeks and then PPB orally administered with HFD for the last 4 weeks (2.5 mg/kg/day).

At the end of the 8-week study period, all mice were sacrificed in accordance with the ethical principles of the Institutional Animal Care and Use Committee of Gachon University (approval number: LCDI-2019-0097).

Point 3: Whilst the pathway for effect of ECE and PPB is proposed via RAGE and TLR4, there is no knockdown experiment to conclusively show this

Response 3: We appreciated with this comment. As your comment, we need to perform the knockdown experiment to accurately show that ECE and PPB attenuated phenotype change of vascular smooth muscle cell to osteoblastic cells and reduced vascular calcification through the RAGE and TLR4 pathways. We agree entirely that the good opinion and some sentences about study limitation are added. However, there are some reference papers that AGE and palmitate acid induce vascular smooth muscle cell phenotype changes and vascular calcification through RAGE and TLR4 signalling pathway including PKC, TGFβ, BMP2, NF-kB, and CBFα-1 [References]. In the further study, we will proceed with the knockdown experiment you mentioned.

References

  1. Kay, A.M.; Simpson, C.L.; Stewart Jr, J.A. The Role of AGE/RAGE Signaling in Diabetes-Mediated Vascular Calcification. Diabetes Res. 2016, 2016, 6809703.
  2. Simard, E.; S¨ollradl, T.; Maltais, J.S.; Boucher, J.; D’Orl´eans-Juste, P.; Grandbois, M. Receptor for advanced glycation endproducts signaling interferes with the vascular smooth muscle cell contractile phenotype and function. PLoS One. 2015, 10, e0128881.
  3. Peng, Y.; Kim, J.-M.; Park, H.-S.; Yang, A.; Islam, C.; Lakatta, E.G.; Lin, L. AGE-RAGE signal generates a specific NF-?B RelA “barcode” that directs collagen I expression. Rep. 2016, 6, 18822.
  4. Chen, N.X.; Duan, D.; O’Neill, K.D.; Moe, S.M. High glucose increases the expression of Cbfa1 and BMP-2 and enhances the calcification of vascular smooth muscle cells. Dial. Transplant. 2006, 21, 3435-3442.
  5. Steitz, S.A.; Speer, M.Y.; Curinga, G.; Yang, H.Y.; Haynes, P.; Aebersold, R.; Schinke, T.; Karsenty, G.; Giachelli, C.M. Smooth muscle cell phenotypic transition associated with calcification: upregulation of Cbfa1 and downregulation of smooth muscle lineage markers. Res. 2001, 89, 1147-1154.

Discussion (previous version)

It is well known that AGE/RAGE pathway is involved in vascular calcification by the enhancement of VSMC phenotype changes to osteoblast-like cells. AGE-treated VSMCs exhibited an increased expression of CBFα-1 mRNA, ALP activity, and osteocalcin secretion [31]. AGEs are synthesized by a reaction between a lysine or a hydroxylysine of a protein and sugar [32]. Therefore, diabetes-induced hyperglycemia is a well-known condition that increases the formation of AGEs [33]. HFD also elevated the AGE deposition. Apart from that, HFD elevated the expressions of RAGE and TLR4 in adipose tissue [26]. In our study, HFD increased the level of serum AGE and expressions of RAGE and TLR4 in mice aorta. It seems that HFD could increase the expressions of RAGE and TLR4 in not only the adipose tissue but also the aorta. Both ECE and PPB reduced the serum level of AGE and expressions of RAGE and TLR4 in the aorta, which was elevated by HFD. We created an in vitro model of HFD by treating palmitate to mouse aortic VSMCs. By treating palmitate, the RAGE and TLR4 expressions were elevated in VSMC, and those were reduced by the addition of either ECE or PPB. AGE treated-VSMC also lead to elevated expressions of RAGE and TLR4. Those expressions were reduced by either ECE or PPB.

Discussion (new version)

It is well known that AGE/RAGE pathway is involved in vascular calcification by the enhancement of VSMC phenotype changes to osteoblast-like cells. AGE-treated VSMCs exhibited an increased expression of CBFα-1 mRNA, ALP activity, and osteocalcin secretion [31]. AGEs are synthesized by a reaction between a lysine or a hydroxylysine of a protein and sugar [32]. Therefore, diabetes-induced hyperglycemia is a well-known condition that increases the formation of AGEs [33]. HFD also elevated the AGE deposition. Apart from that, HFD elevated the expressions of RAGE and TLR4 in adipose tissue [26]. In our study, HFD increased the level of serum AGE and expressions of RAGE and TLR4 in mice aorta. It seems that HFD could increase the expressions of RAGE and TLR4 in not only the adipose tissue but also the aorta. Both ECE and PPB reduced the serum level of AGE and expressions of RAGE and TLR4 in the aorta, which was elevated by HFD. We created an in vitro model of HFD by treating palmitate to mouse aortic VSMCs. By treating palmitate, the RAGE and TLR4 expressions were elevated in VSMC, and those were reduced by the addition of either ECE or PPB. AGE treated-VSMC also lead to elevated expressions of RAGE and TLR4. Those expressions were reduced by either ECE or PPB. Although the knock-down experiments of TLR4 and RAGE were not conducted, we showed that ECE and PPB can inhibit vascular calcification through TLR4 and RAGE through this study.

Reviewer 2 Report

In this interesting paper, the authors explore the mechanisms underlying AGEs/RAGEs and TLR4 induced vascular calcification in an in vivo model of mice on high-fat diet (HFD) and in an in vitro cell culture model employing (murine?) vascular smooth muscle cells (VSMCs).

Analysis in HFD treated mice revealed upregulation of RAGE, TLR4, TGFβ, BMP2, PKC, and NF-κB 20 signals in the aorta of these mice. In contrast, E. cava extract (ECE) or pyrogallol-phloroglucinol-6,6-bieckol (PPB) lowered the induction of those detrimental signalling pathways. In addition, AGE or 21 palmitate-treated VSMCs indicated similar changes as seen in the animal. Furthermore, the authors also observed anti-calcifying protective effects of ECE and PPB on HFD treated mice and AGE or 21 palmitate-treated VSMCs. In consecutive signaling pathway analysis they confirmed the involvement NF-κB/ CBFα-1 pathway.

The experimental approaches are sound and state of the art. In their results section and discussion, the authors delineate a novel mechanism, outlining how ECE and PPB ameliorates VSMC function in AGEs/RAGEs or TLR4 induced vascular calcification with HFD mice and VSMC model. However, there are some considerable flaws in the experimental design and a number of major issues that should be addressed, as outlined in detail below.

Main concerns

  1. In this study muring aortic cells were used instead of human aortic cells. This attenuates the comparability in human situation. Please see discussion of PUBMED-ID (PMID): 29228352 “TNF-alpha in uraemic serum promotes osteoblastic transition and calcification of VSMCs…” and discuss this important aspect in your paper.
  2. To show the consequence of HFD to the mouse the authors should provide a table with key physiological and blood parameters, such as body weight, blood glucose, triglycerides, cholesterol, HDL-cholesterol, LDL-Cholesterol.
  3. How did the authors measure NF-kB mRNA expression since NF-κB transcription factor family in mammals consists of five proteins, p65 (RelA), RelB, c-Rel, p105/p50 (NF-κB1), and p100/52 (NF-κB2) that associate with each other to form distinct transcriptionally active homo- and heterodimeric complexes. It would make more sense to measure the active NF-kB at protein level.
  4. The authors should also mention the purity of their ECE preparation.
  5. The exact number of experiments done with MOVAS should be given.
  6. Since we now that oxidative stress is involved in osteogenic differentiation the authors should measure NOX2 as marker of increased oxidative stress and discuss the molecular mechanism (PMID: 29228352)

Minor concerns

  1. Sentences from 141 to 144 should be rewritten, otherwise it is very hard to understand the procedure of reverse transcription of the RNA into cDNA and following PCR procedure.
  2. In figure 2 column colors at least for the control group should be different to distinguish the different groups
  3. Gene ID for the designing of the indicated primers should be written
  4. Labelling of the columns for the statistic are confusing. The authors may choose asterisks for indicating significance.

Author Response

Response to Reviewer 2 Comments

In this interesting paper, the authors explore the mechanisms underlying AGEs/RAGEs and TLR4 induced vascular calcification in an in vivo model of mice on high-fat diet (HFD) and in an in vitro cell culture model employing (murine?) vascular smooth muscle cells (VSMCs).

Analysis in HFD treated mice revealed upregulation of RAGE, TLR4, TGFβ, BMP2, PKC, and NF-κB 20 signals in the aorta of these mice. In contrast, E. cava extract (ECE) or pyrogallol-phloroglucinol-6,6-bieckol (PPB) lowered the induction of those detrimental signalling pathways. In addition, AGE or 21 palmitate-treated VSMCs indicated similar changes as seen in the animal. Furthermore, the authors also observed anti-calcifying protective effects of ECE and PPB on HFD treated mice and AGE or 21 palmitate-treated VSMCs. In consecutive signaling pathway analysis they confirmed the involvement NF-κB/ CBFα-1 pathway.

The experimental approaches are sound and state of the art. In their results section and discussion, the authors delineate a novel mechanism, outlining how ECE and PPB ameliorates VSMC function in AGEs/RAGEs or TLR4 induced vascular calcification with HFD mice and VSMC model. However, there are some considerable flaws in the experimental design and a number of major issues that should be addressed, as outlined in detail below.

Main concerns

Point 1: In this study muring aortic cells were used instead of human aortic cells. This attenuates the comparability in human situation. Please see discussion of PUBMED-ID (PMID): 29228352 “TNF-alpha in uraemic serum promotes osteoblastic transition and calcification of VSMCs…” and discuss this important aspect in your paper.

Response 1: We appreciate with your comment. As you mentioned, there might be difference between human and mice vascular smooth muscle cells. Thus we added this point in the discussion section of Manuscript.

Discussion (previous version)

In our study, the systolic, diastolic, and mean blood pressures were elevated by HFD, and those are reduced by either PPB or ECE. The intima-media ratio was reduced in the aorta of HFD-fed mice, and it was elevated by either PPB or ECE (Figure 5A-C). It is proposed that the HFD induced the increased medial thickness of the aorta, and it was reduced by treating PPB or ECE. Our results indicated that HFD induced RAGE and TLR4 activation, which leads to the upregulation of TGFβ, BMP2, PKC, and NF-κB signals in the aorta of mice (Figure 3 and sFigure 1). AGE or palmitate treatment in the MOVAS also elevated RAGE and TLR4 expressions and resulted in the upregulation of TGFβ, BMP2, PKC, and NF-κB signals. HFD increased the osteoblast-like VSMCs in the aorta of mice, which expressed CBFα-1 and osteocalcin and manifested an increased ALP activity in the aorta. Vascular calcification, which was evaluated using the calcium deposition ratio and Alizarin red S stain, was increased by HFD. ECD and PPB reduced osteoblast-like VSMCs and vascular calcification in the aorta. PPB and ECE reduced systolic, diastolic, and mean blood pressures, which increased due to HFD. PPB and ECE reduced the phenotype changes to osteoblast-like VSMCs and vascular calcification and therefore decreased the blood pressure (Figure 5F).

Discussion (new version)

In our study, the systolic, diastolic, and mean blood pressures were elevated by HFD, and those are reduced by either PPB or ECE. The intima-media ratio was reduced in the aorta of HFD-fed mice, and it was elevated by either PPB or ECE (Figure 5A-C). It is proposed that the HFD induced the increased medial thickness of the aorta, and it was reduced by treating PPB or ECE. Our results indicated that HFD induced RAGE and TLR4 activation, which leads to the upregulation of TGFβ, BMP2, PKC, NOX2, and NF-κB signals in the aorta of mice (Figure 3 and sFigure 1). AGE or palmitate treatment in the MOVAS also elevated RAGE and TLR4 expressions and resulted in the upregulation of TGFβ, BMP2, PKC, and NF-κB signals. HFD increased the osteoblast-like VSMCs in the aorta of mice, which expressed CBFα-1 and osteocalcin and manifested an increased ALP activity in the aorta. Vascular calcification, which was evaluated using the calcium deposition ratio and Alizarin red S stain, was increased by HFD. ECD and PPB reduced osteoblast-like VSMCs and vascular calcification in the aorta. PPB and ECE reduced systolic, diastolic, and mean blood pressures, which increased due to HFD. PPB and ECE reduced the phenotype changes to osteoblast-like VSMCs and vascular calcification and therefore decreased the blood pressure (Figure 5F).

The limitation of our study is that we used MOVAS instead of human origin VSMC, thus the effect of PPB or ECE on reducing osteoblast-like VSMCs and vascular calcification might be different between human and mice VSMC. In the future study, we should evaluate the of PPB or ECE effect on reducing osteoblast-like VSMCs with human VSMCs for human application. In addition, VSMC could be affected by different underlying comorbidities, it might be important which underlying mechanisms of VSMC calcification in primary isolated human VSMCs from donors of different age groups and with different underlying comorbidities [44].

Point 2: To show the consequence of HFD to the mouse the authors should provide a table with key physiological and blood parameters, such as body weight, blood glucose, triglycerides, cholesterol, HDL-cholesterol, LDL-Cholesterol.

Response 2: We appreciate with your comment. As your comment, HFD-fed mice were used this study and key physiological and blood parameters of the mice were also provided in supplementary figure 1 of manuscript. Body weight increased in HFD mice group than those of NFD mice group but, the body weight in oral administrative ECE or PPB of HFD mice group decreased than those of HFD mice group. As well as body weight of mice, triglyceride, total cholesterol and low density lipoprotein (LDL) level also increased in HFD mice group than those of NFD mice group but, the triglyceride, total cholesterol and low density lipoprotein (LDL) level in oral administrative ECE or PPB of HFD mice group decreased than those of HFD mice group. These information added in results section, supplementary figure 1 and its legend of manuscript

Supplementary figure 1 and legend (new version)

Figure S1. Changes of body weight, triglyceride, total cholesterol and LDL level of HFD-fed mice

C57BL/6N mice (male, 8 weeks of age) were bought and NFD or HFD-fed for 4 weeks and then 0.9% normal saline orally administered with NFD (NFD group) or HFD (HFD gro up) for the last 4 weeks. ECE (ECE group) of PPB (PPB group) orally administered with HFD for the last 4 weeks. At the end of the 8-week study period, (A) body weight measured and then all mice were sacrificed. (B) triglyceride, (C) total cholesterol and (D) LDL level measured in blood serum of mice. The asterisk (*) indicates difference between some groups vs. NFD group and the sharp (#) indicates difference between some groups vs. HFD group. LDL, low-density lipoproteins

Results (previous version)

3.1. ECE and PPB reduced the expression of RAGE and TLR4 increased by HFD in the aorta

The serum level of AGE of HFD-fed mice was higher than in NFD-fed mice, and it was reduced by either PPB or ECE. The reducing effect was most prominent on 150 mg/kg of ECE and PPB administration (Figure 1A). The expression of RAGE in the aorta of HFD-fed mice was increased, and it was reduced by either ECE or PPB administration (Figure 1B). There was no significant difference in the decreasing effect among 50, 100, and 150 mg/kg of ECE and PPB administration. The expression of TLR4 in the aorta of HFD-fed mice was elevated, and it was reduced by PPB or ECE administration (Figure 1C). The reducing effect on TLR4 expression among 100 and 150 mg/kg of ECE and PPB was not significantly different.

Results (new version)

3.1. ECE and PPB reduced the expression of RAGE and TLR4 increased by HFD in the aorta

The C57BL/6N mice were fed with 8-week NFD or HFD and some parameters were validated (Figure S1). The serum level of AGE of HFD-fed mice was higher than in NFD-fed mice, and it was reduced by either PPB or ECE. The reducing effect was most prominent on 150 mg/kg of ECE and PPB administration (Figure 1A). The expression of RAGE in the aorta of HFD-fed mice was increased, and it was reduced by either ECE or PPB administration (Figure 1B). There was no significant difference in the decreasing effect among 50, 100, and 150 mg/kg of ECE and PPB administration. The expression of TLR4 in the aorta of HFD-fed mice was elevated, and it was reduced by PPB or ECE administration (Figure 1C). The reducing effect on TLR4 expression among 100 and 150 mg/kg of ECE and PPB was not significantly different.

Point 3: How did the authors measure NF-kB mRNA expression since NF-κB transcription factor family in mammals consists of five proteins, p65 (RelA), RelB, c-Rel, p105/p50 (NF-κB1), and p100/52 (NF-κB2) that associate with each other to form distinct transcriptionally active homo- and heterodimeric complexes. It would make more sense to measure the active NF-kB at protein level.

Response 3: We appreciate with your comment and agree with your opinion. To measure the active NF-kB at protein level, we performed immunohistochemistry and its method added in material and methods section. The immunohistochemistry result added in figure 3 instead of NF-kB mRNA expression level. Looking at the results, The NF-kB expression level increased in HFD group than those of NFD group but, the expression level decreased in ECE or PPB group than those of HFD group. The protein expression level was quantified using Image J software (NIH) and added the quantified graph in figure 3D.    

Materials and Methods (previous version)

2.4. Immunohistochemistry (3,3-diaminobenzidine; DAB)

Blocks of paraffin-embedded aorta tissue were sectioned to 10 µm thickness, placed on a coating slide, and dried at 37°C for 24-hour. The slides were then deparaffinized with xylene and incubated with 0.3% hydrogen peroxide (Sigma-Aldrich) for 30-minute. Afterward, slides were rinsed twice with PBS and incubated in normal animal serum to reduce non-specific antibody-antigen binding and then incubated with anti-RAGE (Santa Cruz biotechnology, USA; dilution rate 1:200) or anti-TLR4 antibody (Novus Biologicals, USA; dilution rate 1:200) at 4°C for 2-day, followed by three additional rinses with PBS. Slides were then treated with biotinylated secondary antibodies from the ABC kit (Vector Laboratories, dilution rate 1:100), which is incubated for an hour with the blocking solution and rinsed thrice with PBS. Slides were left to react with 3,3′-diaminobenzidine substrates for 15-minute, and they were mounted with a cover slip and DPX mounting solution (Sigma-Aldrich). Images were seen with the use of a light microscope (Olympus, Japan), and the quantification of the intensity of the brown color was made with the use of the Image J software 1.53 version (NIH, USA).

Materials and Methods (new version)

2.4. Immunohistochemistry (3,3-diaminobenzidine; DAB)

Blocks of paraffin-embedded aorta tissue were sectioned to 10 µm thickness, placed on a coating slide, and dried at 37°C for 24-hour. The slides were then deparaffinized with xylene and incubated with 0.3% hydrogen peroxide (Sigma-Aldrich) for 30-minute. Afterward, slides were rinsed twice with PBS and incubated in normal animal serum to reduce non-specific antibody-antigen binding and then incubated with anti-RAGE (Santa Cruz biotechnology, USA; dilution rate 1:200), anti-NF-kB (Cell signaling, USA; dilution rate 1:250) or anti-TLR4 antibody (Novus Biologicals, USA; dilution rate 1:200) at 4°C for 2-day, followed by three additional rinses with PBS. Slides were then treated with biotinylated secondary antibodies from the ABC kit (Vector Laboratories, dilution rate 1:100), which is incubated for an hour with the blocking solution and rinsed thrice with PBS. Slides were left to react with 3,3′-diaminobenzidine substrates for 15-minute, and they were mounted with a cover slip and DPX mounting solution (Sigma-Aldrich). Images were seen with the use of a light microscope (Olympus, Japan), and the quantification of the intensity of the brown color was made with the use of the Image J software 1.53 version (NIH, USA).

Figure 3 and legend (previous version)

Figure 3. Comparative analysis of ECE and PPB treatment on the reduction of vascular smooth muscle cell phenotypic switching to osteoblast-like cell in AGE or palmitate-treated VSMC and HFD-fed mice. mRNA levels of the phenotypic switching to osteoblast-like cell-related molecules A) PCK, B) TGFβ, C) BMP2 and D) NF-κB were determined using qRT-PCR. 25 ug/ml ECE or 1.8 ug/ml PPB were incubated with AGE or PA-treated VSMC (cell) and 100 mg/kg ECE or 2.5 mg/kg PPB administered with HFD-fed mice. Means denoted by a different letter indicate significant differences between groups (p < 0.05).

Figure 3 and legend (new version)

Figure 3. Comparative analysis of ECE and PPB treatment on the reduction of vascular smooth muscle cell phenotypic switching to osteoblast-like cell in AGE or palmitate-treated VSMC and HFD-fed mice. mRNA levels of the phenotypic switching to osteoblast-like cell-related molecules A) PCK, B) TGFβ, C) BMP2 and E) CBFα1 were determined using qRT-PCR. D) Arrows indicate NF-κB protein expression of HFD-fed mice and quantified graph showing intensity. 25 ug/ml ECE or 1.8 ug/ml PPB were incubated with AGE or PA-treated VSMC (cell) and 100 mg/kg ECE or 2.5 mg/kg PPB administered with HFD-fed mice. Means denoted by a different letter indicate significant differences between groups (p < 0.05). scale bar = 25 um

Point 4: The authors should also mention the purity of their ECE preparation.

Response 4: We appreciate with your comment. We prepare the ECE through the following process. Ecklonia cava were thoroughly washed with pure water and air-dried at room temperature about 2 days, the Ecklonia cava were ground, and 50% alcohol was added, followed by incubation at 85°C for 12 hours. The ECE were filtered and then, concentrated with 20% dextrin. The physical amount may be small compared to the efficacy of Ecklonia cava, dextrin is added as an auxiliary to increase the efficiency of granule production. The Ecklonia cava extract were sterilized by heating to high temperature (over 85°C) for 40 to 60-minute, and then spray-dried. Therefore, the purity of ECE is 80%.

Point 5: The exact number of experiments done with MOVAS should be given.

Response 5: We appreciate with your comment. In this study, various experiments were conducted using MOVAS, and experiments were repeatedly conducted at least three times. We added this information to the Materials and methods section to reduce confusion for readers and reviewers.

Material and methods (previous version)

2.3.3. In vitro modeling

To create the palmitate-treated group, 200 uM palmitate with or without 5, 25, or 50 ug/ml ECE or 1.8 ug/ml PPB was treated to MOVAS for 24-hour, and then the cell was washed with phosphate-buffered saline (PBS) thrice. Another way to create the AGE-treated group, 800 ng/ml AGE with or without 5, 25, or 50 ug/ml ECE or 1.8 ug/ml PPB was treated to MOVAS for 24-hour, and then the cell was washed with PBS third time. After all procedures are finished, each experiment was begun.

Material and methods (new version)

2.3.3. In vitro modeling

To create the palmitate-treated group, 200 uM palmitate with or without 5, 25, or 50 ug/ml ECE or 1.8 ug/ml PPB was treated to MOVAS for 24-hour, and then the cell was washed with phosphate-buffered saline (PBS) thrice. Another way to create the AGE-treated group, 800 ng/ml AGE with or without 5, 25, or 50 ug/ml ECE or 1.8 ug/ml PPB was treated to MOVAS for 24-hour, and then the cell was washed with PBS third time. After all procedures are finished, each experiment was begun and all experiments were repeated at least 3 times.

Point 6: Since we now that oxidative stress is involved in osteogenic differentiation the authors should measure NOX2 as marker of increased oxidative stress and discuss the molecular mechanism (PMID: 29228352)

Response 6: We appreciate with your comment. As you recommended, we measured NOX2 in animal model. The NOX2 level increased in HFD group than those of NFD group but, ECE or PPB administration reduced than those of HFD group. So we thought that ECE and PPB treatment could regulate NOX2 expression and this result added as supplementary figure 3 and NOX2 related mechanism in the discussion section.

Supplementary figure 3 and legend (new version) (new)

Figure S3. Inhibitory effects of ECE or PPB administration on NOX2 expression

mRNA levels of NOX2 was determined using qRT-PCR. 50, 100, 150 mg/kg ECE or 2.5 mg/kg PPB were administrated with HFD-fed mice. The asterisk (*) indicates difference between some groups vs. NFD group and the sharp (#) indicates difference between some groups vs. HFD group. The asterisk (*) indicates difference between some groups vs. PBS treated cell group and the sharp (#) indicates difference between some groups vs. AGE or PA treated cell group. NOX2, NADPH oxidase 2

Results (previous version)

3.3 ECE and PPB reduced the expression of PKC, TGFβ, BMP2, and NF-κB by HFD or by either AGE or palmitate

The expressions of PKC, TGFβ, BMP2, and NF-κB in the MOVAS were elevated by either AGE or palmitate. Those expressions were reduced by treating either ECE or PPB. The reducing effect in the 25 and 50 μg/ml of ECE or PPB were not significantly different (Figure 3 and sFigure 1A-H). The expressions of PKC, TGFβ, BMP2, and NF-κB were found in the aorta by HFD, and those were reduced by either PPB or ECE administration. The reducing effect on PKC and TGFβ were most prominent in the PPB-administered mice aorta. The reducing effect on BMP2 was not significantly different between 50 mg/kg of ECE or PPB administration (Figure 3 and sFigure 1I-K). The reducing effect on NF-κB was not significantly different among 50, 100, and 150 mg/kg of ECE and PPB administration (Figure 3 and sFigure 1L).

Results (new version)

3.3. ECE and PPB reduced the expression of PKC, NOX2, TGFβ, BMP2, and NF-κB by HFD or by either AGE or palmitate

The expressions of PKC, TGFβ, BMP2, and NF-κB in the MOVAS were elevated by either AGE or palmitate. Those expressions were reduced by treating either ECE or PPB. The reducing effect in the 25 and 50 μg/ml of ECE or PPB were not significantly different (Figure 3 and sFigure 2A-H).

The increased expressions of PKC, TGFβ, BMP2, and NF-κB were found in the aorta by HFD, and those were reduced by either PPB or ECE administration. The reducing effect on PKC and TGFβ were most prominent in the PPB-administered mice aorta. The reducing effect on BMP2 was not significantly different between 50 mg/kg of ECE or PPB administration (Figure 3 and sFigure 2I-K). The reducing effect on NF-κB was not significantly different among 50, 100, and 150 mg/kg of ECE and PPB administration (Figure 3 and sFigure 2L). The expression of NOX2 was increased by HFD and it was decreased by either of ECE or PPB administration (sFigure 3). The decreasing effect was most prominent in 150 mg/kg of ECE and PPB administration.

Minor concerns

Point 1: Sentences from 141 to 144 should be rewritten, otherwise it is very hard to understand the procedure of reverse transcription of the RNA into cDNA and following PCR procedure.

Response 1: We appreciate with your comment. We have revised the sentences so that readers and reviewers can understand them easily in 2.5. RNA extraction and quantitative real-time polymerase chain reaction (qRT-PCR) of material and methods section.

Material and methods (previous version)

2.5. RNA extraction and quantitative real-time polymerase chain reaction (qRT-PCR)

To extract RNA from cell and aorta tissues, RNAiso Plus (TAKARA, Japan) was used according to the instruction manual. Resuspend the pellets with 1 ml of RNAiso Plus mixed in 0.1 ml of chloroform (Amresco, OH, USA), and then centrifuged at 12,000 x g for 15-minute at 4℃. The clear part (supernatant) was mixed with 0.25 ml absolute isopropanol, and isolated RNA pellets were washed with 70% alcohol and centrifuged at 7,500 x g for 5-minute at 4℃. Dried pellets were dissolved in 5 to 30 ul of diethyl pyrocarbonate treated water and extract RNA was quantified using a Nanodrop 2000 (Thermo Fisher Scientific, MA, USA). Suitable primers diluted with distilled water and the RNA sample from tissue and cells were mixed and put in a 384-well plate, followed by the addition of another 1µg template complementary DNA (cDNA) and SYBR green (TAKARA, Japan). Mixed samples were confirmed with the use of a PCR machine (Bio-Rad, CA, USA). The used primers are outlined in Supplementary Table 1 (Table S1).

Material and methods (new version)

2.5. RNA extraction and quantitative real-time polymerase chain reaction (qRT-PCR)

To extract RNA from cell and aorta tissues, RNAiso Plus (TAKARA, Japan) was used according to the instruction manual. Resuspend the pellets with 1 ml of RNAiso Plus mixed in 0.1 ml of chloroform (Amresco, OH, USA), and then centrifuged at 12,000 x g for 15-minute at 4℃. The clear part (supernatant) was mixed with 0.25 ml absolute isopropanol, and isolated RNA pellets were washed with 70% alcohol and centrifuged at 7,500 x g for 5-minute at 4℃. Dried pellets were dissolved in 5 to 30 ul of diethyl pyrocarbonate (DEPC) treated water and extract RNA was quantified using a Nanodrop 2000 (Thermo Fisher Scientific, MA, USA). Appropriate primers listed in Supplementary Table 1 (Table S1), the forward primers, reverse primers, distilled water, template complementary DNA (cDNA) and SYBR green (TAKARA, Japan) mixed and placed in a 384-well plate. The mixed samples were validated using a PCR machine (Bio-Rad, CA, USA).

Point 2: In figure 2 column colors at least for the control group should be different to distinguish the different groups

Response 2: We appreciate with your comment. As your comment, column colors of all figures were changed to distinguish the different groups. We are confident that this will make it easier for readers and reviewers to understand. In all figures, the column color of the saline orally administrated normal fat diet-fed mice (or PBS treated cell) group is white color, and the column color of the saline orally administrated high fat diet-fed mice (or AGE or PA treated cell) group is black color.

Figures (previous version)

Figure 1

Figure 2

Figure 3

Figure 4

Figure 5

Figures (new version)

Figure 1

Figure 2

Figure 3

Figure 4

Figure 5

Point 3: Gene ID for the designing of the indicated primers should be written

Response 3: We appreciate with nice comment. As your comment, we added Gene ID with primer sequence in supplementary table 1.

Supplementary table 1(new version)

Point 4: Labelling of the columns for the statistic are confusing. The authors may choose asterisks for indicating significance

Response 4: We appreciate with your comment. As your comment, all statistic labelling (a, b, c ···) are changed to special characters (* and #). The asterisk (*) indicates difference between some groups vs. the saline orally administrated normal fat diet-fed mice (or PBS treated cell) group and the sharp (#) indicates difference between some groups vs. the saline orally administrated high fat diet-fed mice (or AGE or PA treated cell) group. We believe that these modifications will make it easier for readers and reviewers to understand significance. This change added in Material and methods section of manuscript and statistical marker of all figures was changed.  

Material and methods (previous version)

 2.9. Statistical analysis

The Kruskal–Wallis and Mann–Whitney U post hoc tests determined the significance of differences among the NFD, HFD, HFD/ECE50, HFD/ECE100, HFD/ECE150, and HFD/PPB groups. Results are presented as mean ± SD, and p-values of <0.05 means it is statistically significant. Means denoted by a different letter indicate significant differences between groups. The analysis was conducted with the use of the SPSS version 22 (IBM Corporation, USA).

Material and methods (new version)

2.9. Statistical analysis

The Kruskal–Wallis and Mann–Whitney U post hoc tests determined the significance of differences among the NFD (PBS), HFD (AGE or PA), HFD/ECE50 (AGE/ECE5 or PA/ECE5), HFD/ECE100 (AGE/ECE25 or PA/ECE25), HFD/ECE150 (AGE/ECE50 or PA/ECE50), and HFD/PPB  (AGE/PPB or PA/PPB) groups. Results are presented as mean ± SD, and p-values of <0.05 means it is statistically significant. Means denoted by a different letter indicate significant differences between groups. The analysis was conducted with the use of the SPSS version 22 (IBM Corporation, USA). The asterisk (*) indicates difference between some groups vs. NFD (or PBS treated cell) group and the sharp (#) indicates difference between some groups vs. HFD (or AGE or PA treated cell) group.

Figures (previous version)

Figure 1

Figure 2

Figure 3

Figure 4

Figure 5

Figures (new version)

Figure 1

Figure 2

Figure 3

Figure 4

Figure 5

Round 2

Reviewer 1 Report

The authors have addressed my concerns.

Reviewer 2 Report

The authors have adequately addressed most, but not all, of the comments raised and in my opinion have significantly improved the manuscript.